# Factors Determining the Mood and Emotions of Nurses Working in Pediatric Wards—A Pilot Study

**DOI:** 10.3390/ijerph20031997

**Published:** 2023-01-21

**Authors:** Anna Bednarek, Krystyna Kowalczuk, Angelika Kucharzyk

**Affiliations:** 1Department of Health Promotion, Faculty of Health Sciences, Medical University of Lublin, 20-059 Lublin, Poland; 2Department of Integrated Medical Care, Faculty of Health Sciences, Medical University of Białystok, 15-089 Białystok, Poland

**Keywords:** mood, emotions, determinants, nurses in pediatric wards, cross-sectional study

## Abstract

Background: Mood and emotions are important aspects of social interactions. In recent years, there has been a growing interest in the participation of these emotional states in the implementation of tasks resulting from specific professions. The aim of the study is to identify the factors that determine the mood and emotions of nurses working in pediatric wards. Methods: The pilot studies presented in the paper were carried out using the diagnostic survey method. The collected data were obtained from the authors’ own questionnaire and the standardized measurement tool “Scale for Measuring Mood and Six Emotions” by Bogdan Wojciszke and Wiesław Baryła. The study included 121 nurses working in hospital pediatric wards. The survey questionnaire results were obtained online using Google Forms. Results: Self-assessment of the health condition of nurses is statistically significantly correlated with all emotions and mood (*p* < 0.05). The better the self-assessment of health, the greater the positive mood measured by the General Mood Scale (GMS) and Mood Scales (MS). The financial situation showed a statistically significant correlation with guilt (*p* = 0.048), sadness (*p* = 0.041), and negative mood (*p* = 0.035). Single people, regardless of gender, were characterized by a greater experience of love (H = 13.497; *p* < 0.001). The higher the education, the greater the experience of love (*p* = 0.009). For people with specialization, the presence of negative emotions such as anger (*p* = 0.039) and guilt (*p* = 0.049) turned out to be statistically significant. The better the health of children staying in the ward, the higher the negative mood of nurses (*p* = 0.035). Conclusions: There was a statistically significant relationship between certain demographic factors, self-assessment of health conditions with the specificity of working in pediatric wards, and the experience of mood and different emotions by nurses working with pediatric inpatients.

## 1. Introduction

Every day, people experience emotions that directly influence their behavior. The occurrence of affects has been developed over the years and is now of a social nature, influencing relations with the immediate environment. Over the centuries, people have also learned to control their emotions [1].

Mood is a concept that expresses the current, subjective emotional state and is characterized by a smooth course and a long duration. A mood can last for hours, days, or even weeks and is only a background for the experienced emotions. Mood usually occurs for no apparent reason and without an external stimulus. Thanks to internal mental processes, a mood is quite regular and stabilized. A balanced (euthymic) mood is characterized by a normal range of fluctuations in the emotional tone [2,3]. On the other hand, a bad mood affects the physical and mental well-being of a person. It manifests itself through experiencing sadness, indifference, and low self-esteem [4,5,6,7]. An elevated mood is caused by various aspects of life and is usually considered an emotional state caused by no apparent reason. The level of intensity of the feelings experienced by a person is above the standard level of intensity of experiencing for that person. Being in a good mood broadens the scope of information selection, thus facilitating rational problem solving [2,3]. In the changing conditions of human life, there are often natural short-term changes of mood consistent with the situation. If the mood swings are much larger and occur without external factors justifying them, there is a need to diagnose and then treat the person experiencing such mood swings [6].

In turn, an emotion is a special type of mental state resulting from the assessment of an event that can significantly influence a person. It is a subjective state because the emotion is felt only by the person affected by the emotion. Emotions are experienced in relation to a person, object, or event. The occurrence of emotions affects the general stimulation of the body and the readiness to take action. It increases muscle tone and motor activity and increases the intensity of mental processes. The autonomic nervous system is also mobilized. The occurrence of emotions can be easily observed in human behavior. Very low emotional arousal manifests itself as apathy, while high emotional arousal is expressed through panic or ecstasy. The duration of emotion can be very different and depends on its type. Being surprised lasts for a short time, while fear can be felt for a long time and can be described in words [8,9].

Emotions are characterized by a specific sign and intensity. The sign classifies emotions into two basic groups—positive and negative. A positive emotion, such as satisfaction or joy, creates a tendency to maintain a certain activity or contact with the object that caused it. A negative emotion, such as anger, fear, or sadness, provokes the interruption of the activity or contact that was the source of the occurrence of the emotion. Negative emotions, despite removing the source that caused them, may still be felt for a certain time [2,4].

The intensity of emotions is an individual matter and it is different for every person. Even the same person may experience their emotions differently depending on the circumstances. Along with the development of personality, a person acquires the ability to control the external symptoms of emotional arousal [1,10]. There are many different classifications of emotions in the literature. One of the most common is the division into basic (primary) and derivative (secondary and complex) emotions. It is assumed that basic emotions, unlike derivative emotions, have characteristic features, such as eliciting a specific reaction, short duration, and quick occurrence [11,12,13,14].

There are a few differences when comparing emotions and mood. The time of experiencing the mood is much longer, while the emotion is temporary and usually lasts a few seconds; only in some cases is it extended to several hours. A mood is less intense. Most often, the mood can be described as positive or negative, or possibly elevated or lowered. When it comes to emotions, it is possible not only to determine whether they are positive or negative but also to determine what emotion is currently being experienced [6,8].

Both mood and emotions are changeable states. Sometimes, an emotion triggered by a given factor can turn into a mood that lasts much longer. There is also the possibility of a situation in which a long-lasting mood turns into emotions [2,3,5].

The nursing profession has a special, multi-tasking nature. It is associated with performing many activities simultaneously or under time pressure, which exposes nurses to various risks related to excessive physical and mental strain [15,16]. It then becomes imperative to have emotional competences. Thanks to these skills, the nurse recognizes and modulates emotions, and uses them when communicating with the patient, the patient’s family, and other members of the medical staff [17,18].

Pediatric nursing is an extremely difficult field that stands out from other specializations. It requires particular responsibility and care for the health of children and youth. The developmental age patient is different from the adult patient. Patients in pediatric wards need treatment, care, and hospitalization adapted to their age and development [19,20,21].

A hospital stay is very often stressful for children due to the limited ability to meet their needs and participate in daily activities typical for their age. This causes an imbalance in the sense of security, which is especially visible in younger children. The young patient usually reacts very emotionally and it is difficult for them to stay calm because they cannot control their emotions yet. The needs of children staying in pediatric wards depend on their age, type and severity of the disease, frequency of hospital stays, development of the nervous system, and their personality. The nurse should first of all provide children with a sense of security by creating an atmosphere of kindness [22,23,24]. The nurse’s care for positive emotional states of children, such as satisfaction, joy, or optimism, has a positive effect on their immune system, regardless of age. Parents are also a big challenge for the staff working in the pediatric ward. They are present with the child whenever possible. As a result of a child’s illness, parents may develop many negative emotions, such as uncertainty, rebellion, a sense of harm, or anger. Under the influence of emotions and stress, they may behave inappropriately towards the staff. In these types of situations, the nurse should remain calm and control negative emotions [25,26,27].

The COVID-19 pandemic turned out to be an extremely difficult time for nurses, including nurses working in pediatric wards. The work of a nurse is stressful, which can result in emotional exhaustion, depersonalization, and reduced personal achievements. Emotional exhaustion is a person’s lack of emotional resources and the feeling that they have nothing more to offer others. Depersonalization develops a negative attitude towards co-workers and patients. These two states are accompanied by the feeling that one’s achievements do not meet personal expectations. Research conducted during the COVID-19 pandemic confirmed that nurses experienced high levels of stress, negative emotions, and occupational burnout. The authors of these studies suggest that healthcare organizations should support nurses by creating internal policies in a given medical facility to protect nurses from occupational burnout. The authors also recommended monitoring nurses for signs and symptoms of burnout and helping them implement strategies to protect their well-being. Psychosocial support combined with self-care training and meditation has been indicated to reduce feelings of insecurity and fear. At the same time, prioritizing rest and breaks when working with patients has proven important. These activities can bring both personal benefits for nurses and the health care system in managing during a pandemic, improving the quality of health care [28,29,30].

In recent years, there has been a growing interest in the role of emotional states in the professional work of medical staff. It is believed that mood and emotions constitute an important aspect of social interactions, and the manner of showing emotions is important not only for patient satisfaction but also contributes to the better functioning of the entire institution providing health services [19]. No studies on the direct impact of mood and emotions on the work of nurses, including pediatric nurses, were found in the Polish and foreign literature. Most often, these issues are discussed to a small extent when discussing other issues—mainly occupational burnout—but are not the main subject of the study. Moreover, the literature does not differentiate between the emotions that occur, but only indicates their negative or positive tone. Meanwhile, the subjectively experienced affect in the form of a non-specific mood or specific emotions is the main thread of human mental life. Experienced emotions and mood constitute both the way of life of an individual, as well as social functioning and the undertaken professional roles of the individual [3,9].

**The aim of this study** is to identify the factors that determine the mood and emotions of the nurses working in pediatric wards.

Mood and emotions motivate our behavior. Thanks to emotions, we establish relationships and function in the work environment. Making nurses more aware of the factors that determine their mood and emotions can help them cope with the demands of their occupational duties, promote their well-being, and promote the quality of nursing care, including pediatric patient safety.

## 2. Materials and Methods

The aim of the study was achieved by conducting a cross-sectional pilot study in a group of 121 pediatric nurses working in a children’s hospital, in treatment and in surgical wards. A cross-sectional study is a type of observational research project where data are obtained over a specific period of time. The aim of the cross-sectional project is to describe the variables and analyze their occurrence and interdependence at a given moment. They are useful for establishing preliminary evidence in planning future advanced studies [31]. This pilot research project aimed to identify factors determining the mood and emotions of nurses working in pediatric wards.

The cross-sectional study presented in the paper was carried out using the diagnostic survey method. The research technique was a questionnaire, and the collected data were obtained from the authors’ own questionnaire and a standardized research tool—the Scale for Measuring Mood and Six Emotions by Bogdan Wojciszke and Wiesław Baryła [32,33]. The above-mentioned research tool consists of the General Mood Scale (GMS), the Positive and Negative Mood Scale (MS), and the Emotion Questionnaire (EQ). They are used both in individual diagnosis and in scientific research.

*The General Mood Scale* (*GMS*) contains 10 statements formulated to show a general positive or negative state. The completion of indicates on a five-point scale how much the respondent agrees or disagrees with each of the statements, according to how the respondent feels at a given moment. The categories are as follows: 1 = I disagree; 2 = I rather disagree; 3 = Kind of yes, kind of no; 4 = I rather agree; 5 = Agree. The result of the scale is the average calculated from individual values.

*Positive and Negative Mood Scales* (*MS*) are adjective lists containing 10 positive and 10 negative adjectives that define the mood. The task of the respondents is to indicate all the adjectives that accurately express their mood during the last week.

*The Emotion Questionnaire* (*EQ*) consists of 24 adjectives that are assigned to one of the six main emotions. According to the questionnaire, the six main emotions are joy, fear, love, guilt, anger, and sadness. The survey consists of the respondent indicating on a seven-point scale how often an emotion is experienced at a certain time. Choosing 1 on the scale means that the given emotion did not take place, while 7 indicates that the given emotion is experienced practically all the time. The different categories of answers are as follows: 1 = never; 2 = very rarely; 3 = rarely; 4 = sometimes; 5 = often; 6 = very often; 7 = always. The score is calculated separately for each of the main emotions by calculating the mean of the six detailed emotion components.

The scales used are characterized by high reliability, and in the case of general mood, very high reliability. Reliability coefficients range from 0.75 to 0.96 when it comes to mood, and from 0.55 to 0.91 when it comes to emotions [29,30].

The own survey questionnaire consisted of 17 closed questions with the possibility of choosing one or more answers. The questionnaire was used to obtain information about the studied group, such as demographic data, social status, professional development, the specificity of working with a pediatric patient, and the conditions of work organization in the ward.

The statistical analysis of the obtained results was prepared in Excel and developed with the use of IBM SPSS Statistics 20 for Windows.

Distributions of quantities and percentages are presented for ordinal and nominal variables, while basic descriptive statistics are presented for quantitative variables. In the analysis of the relationship between the variables, the following tests and statistical coefficients were applied:*The Kolmogorov–Smirnov test,* to establish the compliance of the distribution of the analyzed variables with the normal distribution and to fit the appropriate statistical tests in the analysis.*The Spearman’s rank correlation coefficient,* to establish statistically significant linear correlations between the variables measured at the ordinal or quotient level if the distribution of these variables statistically significantly differs from the normal distribution.*The Mann–Whitney U test,* to determine statistically significant differences between two groups in terms of variables.*The Kruskal–Wallis test,* to show whether more than two groups differ from each other in a statistically significant manner in terms of variables.

The following levels were considered statistically significant: *p* < 0.01, *p* < 0.05, and *p* < 0.001.

### The Course of the Study

The inclusion criteria for the study were as follows:Practicing the nursing profession;Nursing care for a pediatric patient in a hospital system.

The exclusion criteria were as follows:Having the license to practice the nursing profession but not practicing the profession;Nursing care for a pediatric patient in an outpatient system;Nursing care for an adult or elderly patient.

Participants of the study were nurses working in pediatric hospital wards. The results of the survey questionnaire were collected from October 2021 to February 2022 using Google Forms (https://forms.gle/xW7ezNXWzP5Zx4MeA, (accessed on 1 October 2021)). Information about the availability of the online survey for this study was also provided via phone call to the management of the University Children’s Hospital in Lublin and the chairmen of the District Chambers of Nurses and Midwives in Lublin and Kraków. The snowball method was also used, involving the recruitment of pediatric nurses by other nurses. A total of 121 nurses participated in the study. Each of the survey respondents consented to be part of the study and was informed about the anonymity and use of the survey results only for scientific purposes. Conducting an online survey determined the respondents to submit a fully completed questionnaire. The study was conducted after approval issued by the Council of the Faculty of Health Sciences of the Medical University of Lublin (Annex No. 7 to Resolution No. 70/2018–2019 of the Council of the Faculty of Health Sciences of the Medical University of Lublin of 22 October 2020) and in accordance with the assumptions of the Declaration of Helsinki.

The questionnaire was available on the website from 12:00 on 1 October 2021 to 00:00 on 28 February. It took about 40 min to complete the questionnaire. Each nurse who wanted to take part in the study had access to the questionnaire, at any place and time, after logging in to the indicated website and completing the consent form for the study, which was an attachment to the questionnaire. Then, all completed survey questionnaires were carefully reviewed by the authors in order to minimize bias and increase the quality of online surveys [32]. Fully completed questionnaires with consent forms were received from 121 participants. An important part of the questionnaire was standardized research tools. The study was treated as a pilot study.

The research was preceded with the pilot study based on the questionnaire utilized in the authors’ own research. The pilot study encompassed 25 pediatric nurses. The results of the pilot study were included in the analysis of the study group.

## 3. Results

### Characteristics of the Study Group

Among 121 nurses participating in the study, the vast majority (95.9%) were women. Men constituted 4.1% of the respondents. The subjects were from 22 to 59 years old (M = 34.4; SD = 11.25). The respondents had worked as nurses in pediatric wards from 1 to 38 years (M = 11.24; SD = 11.54).

Almost half of the respondents (42.1%) lived in the voivodship city. Nurses residing in the countryside (27.3%), poviat (23.1%), and communal towns (7.4%) were the smallest group. Almost half of the respondents (49.6%) were married, and a slightly smaller percentage (44.6%) was single. A few of the respondents were widowed (3.3%) and divorced (2.5%). A similar percentage of the respondents assessed their financial situation as good (44.6%) and as satisfactory (43.8%). A smaller number of respondents indicated a very good economic status (9.1%), while 2.5% indicated an unsatisfactory financial situation.

Slightly more than half of the respondents (52.1%) completed undergraduate nursing studies and 33.1% completed graduate studies. A smaller group consisted of those who had a medical secondary school certificate (8.3%) or went to a medical vocational college (6.6%). A total of 49 people had a pediatric specialty, and they constituted 40.5% of the respondents. Nearly half of the respondents (49.6%) worked in non-invasive treatment wards of various types, 27.3% in a surgical ward, and 5% in a hospital emergency ward. The remaining 18.2% worked as pediatric nurses in rehabilitation departments, including sanatoriums.

The vast majority of the respondents (85.1%) worked on a 12-h shift. Almost half of the respondents (48.8%) were employed in wards where fewer than three nurses were on duty, and the number of patients ranged from 10 to 20 patients. Every third respondent (34.7%) provided care for children of preschool or school age, and every fourth (24.8%) for infants. Almost half of the respondents (46.3%) worked mainly with children with acute infectious diseases with satisfactory general health condition. Only 5.8% of nurses provided care for children with chronic diseases in very poor general condition. In the opinion of the vast majority of the respondents (84.3%), due to the epidemiological situation caused by COVID-19, work was more difficult. The difference was not noticed by 10.7% of the respondents, and the remaining 5% said that work is now better than before.

Almost half of the respondents (47.1%) assessed their health condition as good. A total of 15.7% of the respondents indicated that their health condition is very good. Slightly fewer respondents (35.5%) assessed their health condition as satisfactory, while only 1.7% assessed their health poorly.

To assess the mood and emotions of the respondents, three tools were used: the General Mood Scale (GMS) consisting of two dimensions, i.e., positive and negative, the Mood Scale (MS), also consisting of two dimensions, i.e., positive and negative, and the Emotion Questionnaire (EQ), consisting of six dimensions, i.e., two positive emotions, joy and love, and four negative emotions, guilt, anger, sadness, and fear. The development of the authors’ own research began with checking the normality of the distributions of the analyzed variables. The Kolmogorov–Smirnov distribution normality test showed that the distributions of seven out of ten analyzed variables were statistically significantly different from the normal distribution. Significant deviations were found only for emotions such as joy, fear, and anger. Therefore, in further analyses, a non-parametric coefficient was used (Table 1).

The descriptive statistics of individual variables for the total number of respondents are presented below, providing the maximum possible range of results in parentheses. The questionnaires do not have standards; therefore, the interpretation of results based on categories such as low, average, and high is not possible.

However, it is possible to check whether the individual personality traits of the respondents and issues related to their work are related to emotions and mood. In the case of ordinal variables, the analyses were performed using the linear correlation coefficient, thus determining whether the higher/lower the level of a given variable leads to a more positive/negative the mood and emotions. On the other hand, in the case of nominal variables, groups of people were compared in terms of emotions and mood. In some nominal variables, groups were combined in such a way as to avoid analyzing small groups separately (Table 2).

No statistically significant differences were found between men and women in terms of mood and emotions.

Age does not significantly correlate with other emotions and moods. The financial situation variable was constructed so that higher categories correspond to a higher status. As it turned out, the better the financial situation, the lower the feeling of guilt and sadness. Additionally, the better the financial situation, the lower the negative mood measured by the GMS. The financial situation does not significantly correlate with other emotions and moods measured by other scales. Self-assessment of health condition statistically significantly correlates with all emotions and moods. The self-assessment of the health condition variable was constructed so that higher categories correspond to better health condition. Hence, the better the health condition, the greater the positive mood and the lower the negative mood measured by both scales. The better the health condition, the greater the joy and love, and the lower the fear, anger, guilt, and sadness. Feelings of guilt and sadness most strongly correlate with a health condition (Table 3).

There were no statistically significant linear correlations between the size of the place of residence and mood and emotions.

On the other hand, marital status statistically significantly differentiates the emotions of the respondents but does not significantly affect their mood. The Single respondents and married respondents are characterized by a significantly greater experience of love than those with a different marital status. In turn, married nurses are characterized by significantly greater fear and guilt than the respondents with other marital statuses (Table 4).

The education variable was constructed so that higher categories correspond to higher education. As it turned out, the higher the education, the greater the experience of love (rho = 0.236; *p* < 0.009**). Education does not significantly correlate with other emotions and mood.

The participants of the study with specialization are characterized by a significantly higher level of anger and guilt than people who do not have specialization. In turn, people without specialization are characterized by significantly greater joy than people with specialization (Table 5).

As shown by the Spearman correlation coefficient (−0.235; *p* < 0.009**), the longer the work experience, the less joy. Work experience does not significantly correlate with other emotions and with positive and negative moods. Similarly, the type of pediatric ward the respondents work in (treatment, surgical, or other types of wards) does not significantly differentiate the respondents in terms of mood and emotions. At the same time, people working 12-h shifts do not statistically differ significantly from people having other working hours in terms of mood and emotions. Moreover, as demonstrated by Spearman’s correlation coefficient (0.278; *p* < 0.002 **), the more shifts a month, the stronger the negative mood measured by Mood Scales. The number of shifts does not significantly correlate with emotions and other mood dimensions.

The most common age of children in a pediatric ward variable was constructed so that the higher the category, the higher the age. As it turned out, the older the children in the ward, i.e., school age, the higher the negative mood measured with the Mood Scales (rho = 0.282; *p* < 0.002 **). The age of children in the ward does not significantly correlate with other mood dimensions and emotions. Additionally, the more nurses on shifts, the greater the anger (rho = −0.181; *p* < 0.046 *). The number of nursing staff working on shifts does not significantly correlate with other emotions and mood scales.

Spearman’s correlation coefficient showed that the more patients in a given pediatric ward, the lower the positive mood measured by the Mood Scales. Additionally, the more patients, the stronger the emotions of fear and sadness. The number of patients does not significantly correlate with other emotions and mood dimensions. The health condition of children staying in the ward variable was constructed so that the better the prognosis of children’s health, the higher the category. Hence, the better the health condition of children in the ward, the higher the negative mood measured by Mood Scales. The health condition of children in the ward does not significantly correlate with other mood dimensions and emotions (Table 6).

The nurses who believe that work during the COVID-19 pandemic is associated with noticeable discomfort are characterized by lower positive mood, measured by both the GMS and the MS, than those who believe working during the pandemic was not more difficult. In addition, people who believe it is not more difficult to work during the pandemic are more joyful than people who believe that work is more difficult. Assessment of the impact of a pandemic on work in the ward does not differentiate negative moods and other emotions (Table 7).

## 4. Discussion

The research conducted in this study was an attempt to determine the factors determining the mood and emotions of the nurses working in pediatric wards.

The available scientific literature on mood and emotions says that a person has the ability to feel many emotions that are triggered by events of everyday life. Through the expression of the body, emotions are made visible and they inform others about the current state of the person. The mood, despite its low intensity, can fluctuate. It is noticed not only by a person who is experiencing it but also by people around them. It can influence the way of interpreting the surrounding world and direct the behavior in interpersonal relationships. On the other hand, emotions play an important role in interpersonal communication, both verbal and non-verbal. Emotions show what a specific person feels and intends to do. Positive emotions of a person, such as joy, bring you closer to them. On the other hand, negative emotions cause reluctance and a tendency to maintain social distance [1,2,31,32,33,34,35,36].

Working in a pediatric ward requires nurses to have a lot of patience, empathy, sensitivity, and cooperation with the family, school, and other institutions. Provided care should be adequate to the developmental age of children and their needs. Professionalism in the work of nurses is very often associated with the ability not to show emotions that are experienced. However, the nature of these emotions significantly affects the physical and mental health of nurses. Too much emotional involvement may contribute to increasing the feeling of negative emotions, contribute to an increase in stress, and lead to occupational burnout [37,38].

Based on the data contained in the questionnaire, the impact of 15 variables constituting factors related to the socio-demographic characteristics of the respondents as well as the environment and work organization on the mood and emotions of pediatric nurses, was analyzed. The indicated variables were: self-assessment of the health of the respondents, sex, age, place of residence, financial situation, marital status, education, including specialty, work experience, age, health condition, number of children in the pediatric ward, number of staff on duty, work on 12-h shifts, work during the COVID-19 pandemic.

The results of the authors’ own research allow to state that the self-assessment of the health condition of the surveyed nurses correlated statistically significantly with all emotions and mood (*p* < 0.05). The better the self-assessment of health condition, the greater the positive mood measured by the GMS and MS. Moreover, the better the health condition, the greater the experience of emotions such as joy and love, and the lower the experience of fear, anger, guilt, and sadness measured with the EQ.

Taking into account the age of the respondents, a statistically significant correlation was noticed only with regard to joy (*p* = 0.015). This leads to the conclusion that the higher the age, the lower the joy. Age does not show a statistically significant correlation with other emotions and mood (*p* > 0.05).

In the present study, the presence of positive mood and emotions was not influenced by the gender of the respondents. Therefore, no statistically significant differences were found with regard to the category of mood and emotions (*p* > 0.05), taking into account the gender of the respondents working as pediatric nurses.

In the authors’ own research, the marital status of the surveyed nurses working in pediatric wards statistically significantly differentiated their emotions. Research shows that married people were more fearful and guilty. On the other hand, single people, regardless of gender, were characterized by experiencing greater love.

The financial situation showed a statistically significant correlation with guilt (*p* = 0.048), sadness (*p* = 0.041), and negative mood (*p* = 0.035). The influence of the financial situation on the higher feeling of positive emotions and lowering the feeling of negative emotions was noticed. However, the higher economic standard of the respondents did not have a statistically significant influence on the occurrence of a positive mood (*p* > 0.05). Similarly, there were no significant correlations between the size of the place of residence and mood and emotions.

When it comes to education, the authors’ own research showed a significant correlation with the positive emotion of love (*p* = 0.009). The higher the education, the greater the experience of the emotions of love. However, in the case of specialization, its influence on the positive emotion of joy (*p* = 0.047), which was characteristic of people without specialization, turned out to be statistically significant. Additionally, in people with specialization, the presence of negative emotions such as anger (*p* = 0.039) and guilt (*p* = 0.049) was found to be statistically significant.

Taking into account the influence of the length of work experience on mood and emotions, this study showed a statistically significant correlation only with joy (*p* = 0.009), which decreased the longer the work experience. This may be due to work fatigue, which in a hospital is associated with high stress resulting from responsibility and constant readiness to perform new tasks.

The impact of the health condition of children staying in a ward on the mood of the nurses is not statistically significant (*p* > 0.05). However, the results of the authors’ own research showed a statistical significance of the correlation with negative mood (*p* = 0.035). The better the health condition of children staying in the ward, the higher the negative mood in the nurses. At the same time, it was shown that the more patients there are in the pediatric ward, the weaker the positive mood measured by the MS and the stronger emotions, such as fear and sadness. The number of patients did not correlate significantly with other emotions and mood dimensions. In addition, the older the children in the pediatric ward, the higher the negative mood measured by the MS. The age of children in the ward did not correlate significantly with other dimensions of mood and emotions.

In the conducted research, nurses working shorter than 12-h shifts did not differ in terms of emotions and mood from nurses working on other shift lengths (*p* > 0.05). Moreover, the number of nurses on duty had a statistically significant influence on one of the emotions studied, namely anger (*p* = 0.046). The more nurses working in a ward on the same shift, the greater the anger. However, no significant correlations with other emotions and mood were found.

This study showed that over 84% of nurses clearly stated that the epidemiological situation caused by the COVID-19 pandemic was associated with noticeable discomfort at work, 5% of the respondents believed that the work is now better than before, and the rest of the respondents did not notice the difference. People who claimed that during the COVID-19 pandemic the work was more difficult were characterized by statistically significantly lower positive mood measured with both the GMS (*p* = 0.009) and the MS (*p* = 0.011). On the other hand, people who believed that the pandemic did not have a negative impact on work were also more joyful than those who said that the work became more difficult (*p* = 0.025).

No sources of similar research in the group of pediatric nurses have been found in the literature. On the other hand, there are studies partially touching on the subject of this paper, which were carried out in a group of nursing staff working in different wards. The scientific literature also provides studies on the influence of certain factors on emotions and mood, which have been carried out on various social groups regardless of age, gender, and profession.

Based on the research conducted by Dziąbek et al. [39], the level of education significantly influenced the occurrence of positive emotions among the nurses. According to the results obtained by the authors of this study, nurses with higher education were characterized by high optimism, which resulted from greater opportunities for social promotion, as opposed to nurses with secondary or vocational education. At the same time, the marital status of the surveyed nurses did not significantly affect the perception of emotions or the ability to express them. Similarly, Marcysiak et al. [40] conducted studies that also show that professional development had a significant impact on experienced emotions. The authors point to the existence of a relationship between education and the perception of negative emotions by people who have not developed professionally. In our study, higher education among pediatric nurses influenced the occurrence of positive emotions, such as love, but having a pediatric specialty was also associated significantly more often with negative emotions, such as anger and guilt. This was probably due to the incorrect system of work organization in the pediatric ward, including the overloading of pediatric nurses with duties, with the simultaneous lack of adequate financial gratification.

Research conducted on a group of 9546 respondents aged 10–85 years by Rutter et al. [41] showed a relationship between age and the feeling of emotions and mood. The authors described the effect of age on decreased feelings of anger and fear, with no decrease in experienced joy. Mature and older people experienced more joy and other positive emotions than young people. Their sensitivity to fear and anger decreased over the years, while their ability to feel and share joy remained the same. Moreover, older people showed more optimism than young people, which may be due to the difference in setting life goals. Mature people and seniors focus their goals primarily on the quality of life “here and now”, and they are able to appreciate the smallest positives in life, such as good relationships with family or friends. Younger people, on the other hand, invest in a better future, often giving up pleasures in the present moment of life. However, our research showed that the older the nurses were, the less joy they were experiencing, which could be due to occupational burnout and incorrect interpersonal relations in the work environment.

The research conducted by Koralewska-Samko and Sadowska [42] showed the differentiation of emotions experienced by children depending on the financial situation of the family. A better financial situation was associated with positive emotions and decreased children’s feelings of fear, anger, and sadness. Similarly, in our research, we showed the importance of better material status when it comes to experiencing positive emotions. However, the higher economic standard of the respondents had no significant effect on the occurrence of positive mood.

In a study of 1200 nurses, Khamisa et al. [43] identified a relationship between work-related stress, occupational burnout, job satisfaction, and the overall health condition of nurses. The results of this study provided empirical evidence that occupational burnout clearly influenced the mental health and well-being of nurses, which was most likely the reason why they experienced negative emotions and moods. In our own research, we also noticed that the better the self-assessment of health in the pediatric nurses was, the greater the experience of emotions such as joy and love, and the lower the feeling of fear, anger, guilt, and sadness.

In the medical literature in recent years, there have been reports reflecting the impact of the COVID-19 pandemic on the work of nurses, such as psychophysical functioning, experiencing stress, experiencing emotions, and occupational burnout.

Galanis et al. [44], based on a systematic review with meta-analysis, evaluated sixteen studies involving 18,935 nurses during the COVID-19 pandemic. They found that nurses experienced high levels of occupational burnout, which was influenced by sociodemographic, social, and occupational factors. The overall prevalence of emotional exhaustion was 34.1%, depersonalization 12.6%, and lack of personal achievements 15.2%. The main risk factors increasing occupational burnout of nurses were younger age, less social support, low readiness of family and colleagues to cope with the COVID-19 pandemic, increased perception of COVID-19-related threats, longer working time in quarantine zones, work in hospitals with inadequate material and human resources, increased workload, and lower levels of specific COVID-19 training.

On the other hand, Sullivan et al. [28] found that nurses working during the COVID-19 pandemic who believed their facility was overstaffed experienced occupational burnout not as often as those who reported that their facility was understaffed. In addition, nurses who felt they were being adequately compensated for their work during the COVID-19 pandemic also experienced less occupational burnout and stress at work.

Weilenmann et al. [30], in a nationwide online survey, assessed the impact of various demographic factors related to work during COVID-19 on the mental health of 1406 healthcare workers, including the prevalence of anxiety, depression, and occupational burnout among the respondents. They found that among healthcare professionals, nurses had the highest levels of anxiety, depression, and occupational burnout, and experienced negative emotions and stress most often. Our study also showed that the vast majority of respondents noticed discomfort at work during the COVID-19 pandemic. At the same time, these people were characterized by a statistically significantly lower positive mood.

Summing up, it should be stated that there is a relationship between certain factors and the experience of various emotions and moods among the nurses working in pediatric wards. A thorough understanding of these dependencies and the factors that determine them may facilitate interpersonal relations and stress management, and improve the quality of medical services provided by nurses.

## 5. Limitations

The authors of this paper hoped that more pediatric nurses would complete the nationwide questionnaire. The number of the respondents, however, may be satisfactory for a pilot study, taking into account the fact that it concerned only nurses working with pediatric patients. The research carried out in this work was a pilot cross-sectional study. Hence, the obtained results allow only general conclusions regarding some aspects of the studied factors, but they do not allow for recognizing the cause-and-effect relationships between them. The obtained results do not fully cover the subject of the influence of many other variables that may condition the mood and emotions of pediatric nurses; therefore, there is a need for further multi-aspect research in this area. In addition, the mood and emotions of pediatric nurses measured in the survey using the aforementioned research tools have a specific local and national context related to the organization of the health care system, conditioned by cultural influences. Therefore, the conclusions of this study cannot be universalized for pediatric nurses in other countries.

## 6. Conclusions

Most of the socio-demographic and work-environment variables analyzed in the study had a significant impact, either positive or negative, on the mood and emotions experienced by the pediatric nurses.Positive mood and emotions in the work of the pediatric nurses were significantly related to factors such as self-assessment of health, age, marital status, financial situation, and education.Variables related to the work environment of pediatric nurses, such as the number and health of children in the ward, workload in a 12-h shift system, the number of staff, and working during the COVID-19 pandemic had a significant impact on the negative mood and emotions experienced by the nurses.Our study proves that some factors cause lower mood and negative emotions in pediatric nurses. This can be a significant problem for hospitals and the healthcare system. Therefore, these factors should be systematically monitored and measures to support the well-being of pediatric nurses should be implemented to ensure quality health care for children.

## Figures and Tables

**Table 1 ijerph-20-01997-t001:** Tests of the normality of the distributions of variables concerning the mood and emotion scales.

Mood and Emotion Scales	Kolmogorov–Smirnov Test		
	K-S	N	*p*
GMS—positive	0.094	121	0.011 *
GMS—negative	0.091	121	0.016 *
MS—positive	0.236	121	<0.001 ***
MS—negative	0.162	121	<0.001 ***
EQ—joy	0.067	121	0.200
EQ—love	0.090	121	0.018 *
EQ—fear	0.069	121	0.200
EQ—anger	0.067	121	0.200
EQ—guilt	0.109	121	0.001 **
EQ—sadness	0.124	121	<0.001 ***

**Legend:** GMS—General Mood Scale; MS—Mood Scales; EQ—Emotion Questionnaire; K-S—statistics of the Kolmogorov–Smirnov test; N—number of cases included; *p*—significance of the Kolmogorov–Smirnov test; ***—the level of statistical significance *p* < 0.001; **—the level of significance *p* < 0.01; *—the level of significance *p* < 0.05.

**Table 2 ijerph-20-01997-t002:** Descriptive statistics of the mood and emotion scales, taking into account all the respondents as a whole.

Mood and Emotion Scales	Min.	Max.	M	Me	SD
GMS—positive (1–5)	1	5	3.37	3.40	0.97
GMS—negative (1–5)	1	5	2.38	2.20	1.02
MS—positive (0–10)	0	8	1.74	1.00	2.04
MS—negative (0–10)	0	9	3.09	3.00	2.10
EQ—joy (1–7)	2.00	7.00	4.49	4.50	1.12
EQ—love (1–7)	2.00	7.00	4.73	4.75	1.16
EQ—fear (1–7)	1.25	7.00	4.02	4.00	1.35
EQ—anger (1–7)	1.00	6.50	3.45	3.50	1.29
EQ—guilt (1–7)	1.00	6.25	2.76	2.50	127
EQ KE—sadness (1–7)	1.00	6.75	2.93	2.75	1.41

**Legend:** GMS—General Mood Scale; MS—Mood Scales; EQ—Emotion Questionnaire; Min—lowest score; Max—highest score; M—arithmetic mean; Me—median; SD—standard deviation.

**Table 3 ijerph-20-01997-t003:** Age, financial situation, and self-assessment of the health of the respondents and the mood and emotions.

Mood and Emotion Scales	Age		Financial Situation		Self-Assessment of Health Condition	
	Rho	*p*	Rho	*p*	Rho	*p*
GMS—positive	0.004	0.967	0.159	0.081	0.326	<0.001 ***
GMS—negative	−0.128	0.163	−0.192	0.035 *	−0.311	0.001 **
MS—positive	−0.167	0.068	0.040	0.659	0.320	<0.001 ***
MS—negative	−0.039	0.673	−0.077	0.404	−0.189	0.038 *
EQ—joy	−0.220	0.015 *	0.131	0.153	0.356	<0.001 ***
EQ—love	−0.039	0.675	0.062	0.497	0.181	0.046 *
EQ—fear	−0.012	0.896	−0.166	0.069	−0.332	<0.001 ***
EQ—anger	0.100	0.277	−0.141	0.122	−0.306	0.001 **
EQ—guilt	0.135	0.140	−0.180	0.048 *	−0.426	<0.001 ***
EQ—sadness	0.034	0.713	−0.186	0.041 *	−0.419	<0.001 ***

**Legend:** GMS—General Mood Scale; MS—Mood Scales; EQ—Emotion Questionnaire; Rho—The Spearman’s rank coefficient of correlation; *p*—the level of statistical significance; ***—the level of statistical significance *p* < 0.001; **—the level of significance *p* < 0.01; *—the level of significance *p* < 0.05.

**Table 4 ijerph-20-01997-t004:** Marital status and the mood and emotions.

Mood and Emotion Scales	Marital Status									Kruskal–Wallis Test	
	Single			Married			Other				
	M	Me	SD	M	Me	SD	M	Me	SD	H	*p*
GMS—positive	3.40	3.40	1.05	3.37	3.60	0.95	3.09	3.20	0.43	1.355	0.508
GMS—negative	2.48	2.50	1.08	2.28	2.10	1.01	2.46	2.40	0.56	1.390	0.499
MS—positive	2.30	2.00	2.38	1.28	1.00	1.55	1.29	0.00	2.21	4.569	0.102
MS—negative	3.04	3.00	2.03	3.17	3.00	2.22	2.86	2.00	1.86	0.063	0.969
EQ—joy	4.66	4.75	1.20	4.37	4.50	1.07	4.14	4.00	0.78	2.675	0.263
EQ—love	4.63	4.75	1.15	5.01	5.00	1.03	3.21	3.00	1.06	13.497	0.001 **
EQ—fear	3.93	3.75	1.29	4.25	4.00	1.36	2.89	2.50	1.16	6.614	0.037 *
EQ—anger	3.28	3.13	1.42	3.65	3.75	1.19	3.07	3.00	1.01	3.266	0.195
EQ—guilt	2.56	2.50	1.23	3.05	3.00	1.26	1.79	1.50	0.92	9.889	0.007 **
EQ—sadness	2.91	2.50	1.42	3.09	3.00	1.42	1.79	2.00	0.74	5.801	0.055

**Legend:** GMS—General Mood Scale; MS—Mood Scales; EQ—Emotion Questionnaire; M—arithmetic mean; Me—median; SD—standard deviation; H—statistics of the Kruskal–Wallis test; *p*—significance of the Kruskal–Wallis test; **—the level of significance *p* < 0.01; *—the level of significance *p* < 0.05.

**Table 5 ijerph-20-01997-t005:** Having a specialization and the mood and emotions.

Mood and Emotion Scales	Specialization						The Mann–Whitney U Test	
	Yes			No				
	M	Me	SD	M	Me	SD	Z	*p*
GMS—positive	3.29	3.40	0.97	3.41	3.40	0.98	−0.501	0.617
GMS—negative	2.34	2.20	1.04	2.41	2.40	1.01	−0.352	0.725
MS—positive	1.29	1.00	1.66	2.04	1.00	2.23	−1.588	0.112
MS—negative	3.22	3.00	2.17	3.00	3.00	2.06	−0.627	0.531
EQ—joy	4.23	4.25	1.10	4.66	4.75	1.11	−1.988	0.047 *
EQ—love	4.73	4.75	1.25	4.74	4.75	1.11	−0.246	0.806
EQ—fear	4.02	4.00	1.34	4.03	4.00	1.36	−0.048	0.962
EQ—anger	3.73	3.75	1.11	3.27	3.25	1.38	−2.068	0.039 *
EQ—guilt	2.97	3.00	1.19	2.61	2.38	1.30	−1.972	0.049 *
EQ—sadness	2.99	2.75	1.37	2.90	2.50	1.44	−0.513	0.608

**Legend:** GMS—General Mood Scale; MS—Mood Scales; EQ—Emotion Questionnaire; M—arithmetic mean; Me—median; SD—standard deviation; Z—statistics for the Mann–Whitney U test; *p*—significance of the Mann–Whitney U test; *—the level of significance *p* < 0.05.

**Table 6 ijerph-20-01997-t006:** The number of patients in the pediatric ward and their health condition, and the mood and emotions.

Mood and Emotion Scales	Number of Patients in the Ward		Health Condition of Children in the Ward	
	Rho	*p*	Rho	*p*
GMS—positive	−0.097	0.288	−0.086	0.351
GMS—negative	0.160	0.080	0.052	0.575
MS—positive	−0.199	0.029 *	0.017	0.852
MS—negative	0.075	0.414	0.192	0.035 *
EQ—joy	−0.126	0.169	−0.148	0.106
EQ—love	−0.022	0.807	−0.033	0.722
EQ—fear	0.207	0.023 *	0.096	0.293
EQ—anger	0.099	0.280	0.073	0.425
EQ—guilt	0.057	0.531	0.077	0.404
EQ—sadness	0.185	0.042 *	0.138	0.132

**Legend:** GMS—General Mood Scale; MS—Mood Scales; EQ—Emotion Questionnaire; Rho—Spearman’s correlation coefficient; *p*—the level of statistical significance; *—the level of significance *p* < 0.05.

**Table 7 ijerph-20-01997-t007:** Assessment of the impact of the COVID-19 pandemic on the work in the ward and the mood and emotions.

Mood and Emotion Scales	The Impact of the Epidemic on Work in the Ward						The Mann–Whitney U Test	
	Work is More Difficult			No Difference of Better				
	M	Me	SD	M	Me	SD	Z	*p*
GMS—positive	3.28	3.40	0.97	3.84	4.00	0.88	−2.612	0.009 **
GMS—negative	2.42	2.40	1.03	2.13	2.00	0.93	−1.147	0.252
MS—positive	1.48	1.00	1.82	3.11	3.00	2.62	−2.536	0.011 *
MS—negative	3.17	3.00	2.17	2.68	3.00	1.67	−0.582	0.561
EQ—joy	4.38	4.50	1.11	5.03	5.25	1.06	−2.239	0.025 *
EQ—love	4.64	4.75	1.14	5.22	5.50	1.17	−1.936	0.053
EQ—fear	4.06	4.00	1.34	3.83	3.75	1.40	−0.746	0.456
EQ—anger	3.50	3.50	1.28	3.21	3.00	1.38	−0.938	0.348
EQ—guilt	2.78	2.50	1.31	2.63	2.75	0.99	−0.164	0.869
EQ—sadness	3.01	2.75	1.41	2.51	2.25	1.36	−1.471	0.141

**Legend:** GMS—General Mood Scale; MS—Mood Scales; EQ—Emotion Questionnaire; M—arithmetic mean; Me—median; SD—standard deviation; Z—statistics for the Mann–Whitney U test; *p*—significance of the Mann–Whitney U test. **—the level of significance *p* < 0.01; *—the level of significance *p* < 0.05.

## Data Availability

The datasets used and/or analyzed during the current study are available from the corresponding author on request.

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
