# Peer review of "Factors Determining the Mood and Emotions of Nurses Working in Pediatric Wards—A Pilot Study"

_ijerph, 2023, doi:10.3390/ijerph20031997_

Round 1

Reviewer 1 Report

This is an interesting manuscript. The article needs editing on English grammar/punctuation organization, include more recent research and major changes. See other comments below.

Abstract

1. Authors say: “The aim of this study is to try to identify”. Please, you have to indicate the action clearly. Better indicate “The aim of this study:  to identify…”.  

2. Keywords: include “cross-sectional”.

Introduction and Background:

1.- Check the information sources of the entire article. There are some works that have been updated and show more current data. Data published in 2002, 2000, 2006, etc. Please, searches must be up to date. This section has to be reworked by adding current publications. I suggest including more recent research that examines the problem.

2.- Please give specific data on the study area. Current actions or protocols that are being carried out in hospitals in the study area. How do institutions help nurses to do better emotional management? Add these ideas that help the reader to contextualize the problem.

3.- Try to balance the information. 70% of the section shows data in relation to the emotional perspective description. Very few data are provided from the perspective of nurses, clinics and health staff.

4.- No mention about Covid-19 pandemic. How the pandemic influenced the emotional experience of nurses

Aim:

Line 158: Please, remove “try”.

Materials/Methods:

Line 161: you say “The study presented in the paper was carried out using the diagnostic survey method. The research technique was a questionnaire”. Please, first define design, and then explain population/sample. Finally, collection instrument. This section needs to be reorganized. Information is missing.

Please define cross-sectional method using different sources.

How was the sample calculation done?

Participants and sampling method: How was the study presented to the participants? Explain the process of how the participants knew about the existence of this study. More information is needed to explain how the study is presented to the participants: Was it in a face-to-face, by phone, by e-mail? How do the researches access to the personal data of the nurses? Were they (participants) able to read the informed consent carefully? Where and when did the nurse complete the questionnaire? Include approximate response time. Please, add this information and give more details about informant consent in this section.

Need to give more detail on inclusion/exclusion criteria. What type of emotional or physical health situations contra-indicated study participation? Please explain this process in detail.

Did a pilot test take place? How was the content of the questionnaire validated? This important information is not added. Please explain.

Type of sampling. Cite source.

No mention about how when the information is obtained (start and end date).  Give details about that, please.

Ethical consideration: Please give details about ethical codes. Was the approval of the ethics committee of the hospitals obtained? Include the approval code in this section.

Characteristics of the studied group… is it part of the results? Please, check it. This information may appear in results section.

Results

This section is well done. The results could be presented in a more clarifying way, but it is correct. Well done!

Discussion

The discussion does not correspond to the extent of the results or the topics covered in it. The discussion is much more reduced than the results obtained. Please check it.

Some reorganization is required. I suggest writing the discussion in more order, giving the information according to the stated results (all of them).

Include a section on Implications for Research/practice.

This section is outdated. Please, review the background maximum 5 years ago.

Strengths and limitations

If the sample is not representative, the results are truly extrapolable and valid? Please reflect on them.

Conclusion

The conclusions are focused on the implications for practice. I suggest focusing the conclusion on the main ideas that the study brings to the world. For example, make a mention of a comprehensive action that can help with the emotional management. Be sure to mention exactly what is your study contribution. What does your study contribute in the clinical context and for the nursing discipline?

Author Response

The authors of the article would like to thank all Reviewers for their valuable comments. The table below shows changes to the content of the article based on the suggestions provided.

Reviewer's recommendations

Authors’ Responses (highlighted in yellow)

Abstract

1. Authors say: “The aim of this study is to try to identify”. Please, you have to indicate the action clearly. Better indicate “The aim of this study:  to identify…”.  

Abstract is now: „The aim of the study is to identify the factors determining the mood and emotions of nurses working in pediatric wards”.

2. Keywords: include “cross-sectional”.

Keywords are now: mood; emotions; determinants; nurses in pediatric wards; cross-sectional

Introduction and Background:

1.- Check the information sources of the entire article. There are some works that have been updated and show more current data. Data published in 2002, 2000, 2006, etc. Please, searches must be up to date. This section has to be reworked by adding current publications. I suggest including more recent research that examines the problem.

2.- Please give specific data on the study area. Current actions or protocols that are being carried out in hospitals in the study area. How do institutions help nurses to do better emotional management? Add these ideas that help the reader to contextualize the problem.

3.- Try to balance the information. 70% of the section shows data in relation to the emotional perspective description. Very few data are provided from the perspective of nurses, clinics and health staff.

4.- No mention about Covid-19 pandemic. How the pandemic influenced the emotional experience of nurses

Text added: The COVID-19 pandemic turned out to be an extremely difficult time for nurses, including nurses working in pediatric wards. The work of a nurse is stressful, which can result in emotional exhaustion, depersonalization, and reduced personal achievements. Emotional exhaustion is a person's lack of emotional resources and the feeling that they have nothing more to offer others. Depersonalization develops a negative attitude towards co-workers and patients. These two states are accompanied by the feeling that one's achievements do not meet personal expectations. Research conducted during the COVID-19 pandemic confirmed that nurses experienced high levels of stress, negative emotions, and occupational burnout. The authors of these studies suggest that healthcare organizations should support nurses by creating internal policies in a given medical facility to protect nurses from occupational burnout. The authors also recommend monitoring nurses for signs and symptoms of burnout and helping them implement strategies to protect their well-being. Psychosocial support combined with self-care training and meditation has been indicated to reduce feelings of insecurity and fear. At the same time, prioritizing rest and breaks when working with patients has proven important. These activities can bring both personal benefits for nurses and the health care system in managing during a pandemic, improving the quality of health care.

References:

28. Sullivan, D.; Sullivan, V.; Weatherspoon, D.; Frazer, Ch. Comparison of Nurse Burnout, Before and During the COVID-19 Pandemic. Nurs Clin North Am 2022, 57(1), 79–99.

29. Tunaiji, H.AL.; Qubaisi, M.Al.; Dalkilinc, M.; Campos, L.A.; Ugwuoke, N.V.; Alefishat, E.; Aloum, L.; Ross, R.; Almahmeed, W.; Baltatu, O.C. Impact of COVID-19 Pandemic Burnout on Cardiovascular Risk in Healthcare Professionals Study Protocol: A Multicenter Exploratory Longitudinal Study. Front Med (Lausanne) 2020, 7, 571057.

30. Weilenmann, S.; Ernst, J.; Petry, H.; Pfaltz, M.C.; Sazpinar, O.; Gehrke, S.; Paolercio, F.; von Känel, R.; Spiller, T.R. Health Care Workers' Mental Health During the First Weeks of the SARS-CoV-2 Pandemic in Switzerland - A Cross-Sectional Study. Front Psychol 2021, 12p. 594340.

Aim: Line 158: Please, remove “try”.

Aim is now: “The aim of this study is to identify the factors that determine the mood and emotions of nurses working in pediatric wards”.

Materials/Methods:

Line 161: you say “The study presented in the paper was carried out using the diagnostic survey method. The research technique was a questionnaire”. Please, first define design, and then explain population/sample. Finally, collection instrument. This section needs to be reorganized. Information is missing.

Please define cross-sectional method using different sources.

How was the sample calculation done?

Text added: The aim of the study was achieved by conducting a cross-sectional pilot study in a group of 121 pediatric nurses working in children's hospital, treatment, and surgical wards. A cross-sectional study is a type of observational research project where data is obtained over a specific period of time. The aim of the cross-sectional project is to describe the variables and analyze their occurrence and interdependence at a given moment. They are useful for establishing preliminary evidence in planning future advanced studies [31]. This pilot research project aimed to identify factors determining the mood and emotions of nurses working in pediatric wards.

References:

31. Wang, X.;  Cheng, Z. Cross-Sectional Studies: Strengths, Weaknesses, and Recommendations. Chest 2020, 158(1S), S65-S71.

Participants and sampling method: How was the study presented to the participants? Explain the process of how the participants knew about the existence of this study. More information is needed to explain how the study is presented to the participants: Was it in a face-to-face, by phone, by e-mail? How do the researches access to the personal data of the nurses? Were they (participants) able to read the informed consent carefully? Where and when did the nurse complete the questionnaire? Include approximate response time. Please, add this information and give more details about informant consent in this section.

Need to give more detail on inclusion/exclusion criteria. What type of emotional or physical health situations contra-indicated study participation? Please explain this process in detail.

Did a pilot test take place? How was the content of the questionnaire validated? This important information is not added. Please explain.

Type of sampling. Cite source.

No mention about how when the information is obtained (start and end date).  Give details about that, please.

Text added: Information about the availability of the online survey for this study was also provided via phone call to the management of the University Children's Hospital in Lublin and the chairmen of the District Chambers of Nurses and Midwives in Lublin and Kraków. The snowball method was also used, involving the recruitment of pediatric nurses by other nurses.

Text added: The questionnaire was available on the website from 12.00 on October 1, 2021, to 00.00 on February 28. It took about 40 minutes to complete the questionnaire. Each nurse who wanted to take part in the study had access to the questionnaire, at any place and time, after logging in to the indicated website and completing the consent form for the study, which was an attachment to the questionnaire. Then, all completed survey questionnaires were carefully reviewed by the authors in order to minimize bias and increase the quality of online surveys [32]. Fully completed questionnaires with consent forms were received from 121 participants. An important part of the questionnaire was standardized research tools. The study was treated as a pilot study.

References:

32. Ball, H.J. Conducting Online Surveys. J Hum Lact 2019, 35(3), 413-417.

Explanation for the Reviewer: There were no other inclusion/exclusion criteria for respondents to participate in the study other than those listed in section 2.1 The course of the study.

Ethical consideration: Please give details about ethical codes. Was the approval of the ethics committee of the hospitals obtained? Include the approval code in this section.

Characteristics of the studied group… is it part of the results? Please, check it. This information may appear in results section.

Explanation for the Reviewer: This is an online survey in which nurses from all over Poland could participate. Only nurses who wanted to complete the questionnaire were asked for their consent to participate in the study. The characteristics of the study group are part of the results, so it has been moved to the results section.

Results: This section is well done. The results could be presented in a more clarifying way, but it is correct. Well done!

Thank you very much to the Reviewer.

Discussion: The discussion does not correspond to the extent of the results or the topics covered in it. The discussion is much more reduced than the results obtained. Please check it.

Some reorganization is required. I suggest writing the discussion in more order, giving the information according to the stated results (all of them).

Include a section on Implications for Research/practice.

This section is outdated. Please, review the background maximum 5 years ago.

Text added: Based on the data contained in the questionnaire, the impact of 15 variables constituting factors related to the socio-demographic characteristics of the respondents as well as the environment and work organization on the mood and emotions of pediatric nurses was analyzed. The indicated variables were: self-assessment of the health of the respondents, sex, age, place of residence, financial situation, marital status, education, including specialty, work experience, age, health condition and number of children in the pediatric ward, number of staff on duty, work on 12-hour shifts, work during the COVID-19 pandemic.

Text added: Similarly, there were no significant correlations between the size of the place of residence and mood and emotions.

Text added: At the same time, it was shown that the more patients there are in the pediatric ward, the weaker the positive mood measured by Mood Scales and the stronger emotions such as fear and sadness. The number of patients did not correlate significantly with other emotions and mood dimensions. In addition, the older the children in the pediatric ward, the higher the negative mood measured by the Mood Scales. The age of children in the ward did not correlate significantly with other dimensions of mood and emotions.

Text added: In our study, higher education of pediatric nurses influenced the occurrence of positive emotions such as love, but having a pediatric specialty was also associated significantly more often with negative emotions such as anger and guilt. This was probably due to the incorrect system of work organization in the pediatric ward, including the overloading of pediatric nurses with duties, with the simultaneous lack of adequate financial gratification.

Text added: However, our research showed that the older the nurses were, the less joy they were experiencing, which could be due to occupational burnout and incorrect interpersonal relations in the work environment.

Text added: Similarly, in our research, we showed the importance of better material status when it comes to experiencing positive emotions. However, the higher economic standard of the respondents had no significant effect on the occurrence of positive mood.

Text added: In our own research, we also noticed that the better the self-assessment of health by pediatric nurses was, the greater the experience of such emotions as joy and love, and the lower the feeling of fear, anger, guilt and sadness.

Text added: In the medical literature in recent years, there have been reports reflecting the impact of the COVID-19 pandemic on the work of nurses, such as psychophysical functioning, experiencing stress, experiencing emotions, and occupational burnout.

Galanis et al. [44], based on a systematic review with meta-analysis, evaluated sixteen studies involving 18 935 nurses during the COVID-19 pandemic. They found that nurses experienced high levels of occupational burnout, which was influenced by sociodemographic, social, and occupational factors. The overall prevalence of emotional exhaustion was 34.1%, depersonalization 12.6%, and lack of personal achievements 15.2%. The main risk factors increasing occupational burnout of nurses were younger age, less social support, low readiness of family and colleagues to cope with the COVID-19 pandemic, increased perception of COVID-19-related threats, longer working time in quarantine zones, work in hospitals with inadequate material and human resources, increased workload and lower levels of specific COVID-19 training.

On the other hand, Sullivan et al. [28] found that nurses working during the COVID-19 pandemic who believed their facility was overstaffed experienced occupational burnout not as often as those who reported that their facility was understaffed. In addition, nurses who felt they were being adequately compensated for their work during the COVID-19 pandemic also experienced less occupational burnout and stress at work.

Weilenmann et al. [30] in a nationwide online survey assessed the impact of various demographic factors related to work during COVID-19 on the mental health of 1406 healthcare workers, including the prevalence of anxiety, depression, and occupational burnout among the respondents. They found that among healthcare professionals, nurses had the highest levels of anxiety, depression, and occupational burnout, and experienced negative emotions and stress most often. Our study also showed that the vast majority of respondents noticed discomfort at work during the COVID-19 pandemic. At the same time, these people were characterized by a statistically significantly lower positive mood.

References:

28. Sullivan, D.; Sullivan, V.; Weatherspoon, D.; Frazer, Ch. Comparison of Nurse Burnout, Before and During the COVID-19 Pandemic. Nurs Clin North Am 2022, 57(1), 79–99.

30. Weilenmann, S.; Ernst, J.; Petry, H.; Pfaltz, M.C.; Sazpinar, O.; Gehrke, S.; Paolercio, F.; von Känel, R.; Spiller, T.R. Health Care Workers' Mental Health During the First Weeks of the SARS-CoV-2 Pandemic in Switzerland - A Cross-Sectional Study. Front Psychol 2021, 12p. 594340.

44. Galanis, P.; Vraka, I.; Fragkou, D.; Bilali, A.; Kaitelidou, D.  Nurses' burnout and associated risk factors during the COVID-19 pandemic: A systematic review and meta-analysis. J Adv Nurs 2021,77(8), 3286-3302.

Strengths and limitations: If the sample is not representative, the results are truly extrapolable and valid? Please reflect on them.

Text added: The authors of this paper hoped that more pediatric nurses would complete the nationwide questionnaire. The number of respondents, however, may be satisfactory for the pilot study, taking into account the fact that it concerned only nurses working with pediatric patients. The research carried out in this work was a pilot cross-sectional study. Hence, the obtained results allow only general conclusions regarding some aspects of the studied factors, but they do not allow for recognizing the cause-and-effect relationships between them. The obtained results do not fully cover the subject of the influence of many other variables that may condition the mood and emotions of pediatric nurses, therefore there is a need for further multi-aspect research in this area.

Conclusion: The conclusions are focused on the implications for practice. I suggest focusing the conclusion on the main ideas that the study brings to the world. For example, make a mention of a comprehensive action that can help with the emotional management. Be sure to mention exactly what is your study contribution. What does your study contribute in the clinical context and for the nursing discipline?

The content of the conclusions has been modified to:

1.         Most of the socio-demographic and work-environment variables analyzed in the study had a significant impact, either positive or negative, on the mood and emotions experienced by pediatric nurses.

2.         Positive mood and emotions in the work of pediatric nurses were significantly related to such factors as self-assessment of health, age, marital status, financial situation and education. 

3.         Variables related to the work environment of pediatric nurses, such as the number and health of children in the ward, workload in the 12-hour shifts system, the number of staff, and working during the COVID-19 pandemic had a significant impact on the negative mood experienced by nurses and emotions. 

4.         Our study proves that some factors cause lower mood and negative emotions in pediatric nurses. This can be a significant problem for hospitals and the healthcare system. Therefore, these factors should be systematically monitored and measures to support the well-being of pediatric nurses should be implemented to ensure quality health care for children.

Reviewer 2 Report

This paper conducted a pilot study to identify the factors which determine the mood and emotions of nurses working in pediatric wards. It suits for the Journal and the Special Issue. However, I think a couple of major issues need to be addressed to improve it before publication.

(1) In "1. Introduction", almost all the text was used to introduce "mood and emotion", and it was explained too much in detail, which should be simplified.

(2) In "1. Introduction", "No studies have been found in Polish and foreign literature on the direct influence of mood and emotions on the work of pediatric nurses", however, the references related to nurses, not only pediatric nurses, should also be reviewed.

(3) In "2. Materials and methods", a total of 121 nurses were surveyed, the rationality of the sample number should be explained, the number of the sample could be got according to the method provided by Czaja and Blair (2005).

Ref list:

Czaja, R., Blair, J., 2005. Designing Surveys: A Guide to Decisions and Procedures. Sage Publications, Inc, United States of America.

S.M.H. Al-Tmeemy, H. Abdul-Rahman, Z. Harun, Contractors' perception of the use of costs of quality system in Malaysian building construction projects, Int. J. Proj. Manag. 30 (7) (2012) 827–838

(4) In "2.2. Characteristics of the studied group", a Table or a Fig. should be given to show the information of the participants.

(5) The Implications of the work should be given.

(6) In "6. Conclusions", the applicability of the findings/results should be given, and this part should be rewritten.

Author Response

REVIEW NO. 2 (highlighted in blue)

Reviewer's recommendations

Authors’ Responses

This paper conducted a pilot study to identify the factors which determine the mood and emotions of nurses working in pediatric wards. It suits for the Journal and the Special Issue. However, I think a couple of major issues need to be addressed to improve it before publication.

(1) In "1. Introduction", almost all the text was used to introduce "mood and emotion", and it was explained too much in detail, which should be simplified.

We agree with the Reviewer and also due to a similar suggestion of the third Reviewer, after thinking about it, we modify the text by cutting it in three places - highlighted in green.

(2) In "1. Introduction", "No studies have been found in Polish and foreign literature on the direct influence of mood and emotions on the work of pediatric nurses", however, the references related to nurses, not only pediatric nurses, should also be reviewed.

We agree with the Reviewer and propose to change the text to: „No studies on the direct impact of mood and emotions on the work of nurses, including pediatric nurses, were found in the Polish and foreign literature”.

(3) In "2. Materials and methods", a total of 121 nurses were surveyed, the rationality of the sample number should be explained, the number of the sample could be got according to the method provided by Czaja and Blair (2005).

Ref list:

Czaja, R., Blair, J., 2005. Designing Surveys: A Guide to Decisions and Procedures. Sage Publications, Inc, United States of America.

S.M.H. Al-Tmeemy, H. Abdul-Rahman, Z. Harun, Contractors' perception of the use of costs of quality system in Malaysian building construction projects, Int. J. Proj. Manag. 30 (7) (2012) 827–838

Explanation for the Reviewer: Only that many people completed the nationwide online survey at the scheduled time. Explanatory text highlighted in yellow has been added to the Materials and methods section.

The added text is as follows (highlighted in yellow): The questionnaire was available on the website from 12.00 on October 1, 2021, to 00.00 on February 28. It took about 40 minutes to complete the questionnaire. Each nurse who wanted to take part in the study had access to the questionnaire, at any place and time, after logging in to the indicated website and completing the consent form for the study, which was an attachment to the questionnaire. Then, all completed survey questionnaires were carefully reviewed by the authors in order to minimize bias and increase the quality of online surveys [32]. Fully completed questionnaires with consent forms were received from 121 participants. An important part of the questionnaire was standardized research tools. The study was treated as a pilot study.

(4) In "2.2. Characteristics of the studied group", a Table or a Fig. should be given to show the information of the participants.

Explanation for the Reviewer: Data from the "Characteristics of the studied group" section have been moved to the "Results" chapter as suggested by the first Reviewer. They contain all the data that was included in the survey questionnaire on the study group of pediatric nurses. Since the paper already contains 7 tables, the authors of the paper propose to leave this data in the form of a description.

(5) The Implications of the work should be given.

(6) In "6. Conclusions", the applicability of the findings/results should be given, and this part should be rewritten.

Explanation for the Reviewer: The implications for the work and the application of the conclusions are included in the following modification of the text of the conclusions (highlighted in yellow):

1.         Most of the socio-demographic and work-environment variables analyzed in the study had a significant impact, either positive or negative, on the mood and emotions experienced by pediatric nurses.

2.         Positive mood and emotions in the work of pediatric nurses were significantly related to such factors as self-assessment of health, age, marital status, financial situation, and education. 

3.         Variables related to the work environment of pediatric nurses, such as the number and health of children in the ward, workload in the 12-hour shifts system, the number of staff, and working during the COVID-19 pandemic had a significant impact on the negative mood experienced by nurses and emotions. 

4.         Our study proves that some factors cause lower mood and negative emotions in pediatric nurses. This can be a significant problem for hospitals and the healthcare system. Therefore, these factors should be systematically monitored and measures to support the well-being of pediatric nurses should be implemented to ensure quality health care for children.

Reviewer 3 Report

Dear Authors: 

Thanks by share your study. 

I have some observations. 

1. Introduction is extensive. There is an explain about the definition of mood, including the types. But all thats concepts are not  used in the rest of the paper, so I suggest summarizing.

The same goes for concept of emotion. For example, you present Plutchik's theory of emotions, where the basic emotions are listed, including disgust and trust. These emotions are not included in this way in the  instrument of evaluation that you later use. 

2. The aim of the study is very broad. As a pilot study, I'm agree using this kind of objective. But you can give  to the readers a stronger arguments in the introduction about why you want to work in this topic. I suggest that clarify if  your focus is on engage, or work environment, well being, patient safety, second victim, or another. 

3. When you use a methodology of investigation an instrument, it's good to explain why did you chosen a survey to explore the mood and emotions; and then, why did you chosen the Scale for measurements of mood and six emotions. Are there another methodologies and instruments? Why are the ones you chose correct?

4. In line 184, when you explained how to get score of emotion questionnaire, wrote "by calculating the mean of the four detailed emotion components". Is not clear, because the instrument contain six emotions. 

5. You obtained a lot of results, and you mentioned the relation of some variables selected with general mood scale, positive and negative, positive and negative mood scales and six emotions. Some results are surprising and need to give to readers some hypothesis that you can try to advance in discussion item. For example, why the people with specialization present more anger and guilty. Unless the sample was so small on this specific topic that the finding isn't really relevant. But this should also clarify it.

6.I suggest including in the limitations a mention that the mood and emotions measured through a survey are affected by a specific local context (my service, my hospital) and national (my health system, my country, context), contingencies, and also cultural influences. Thus, the conclusions of this study cannot be universalized to other pediatric nurses in other countries.

Again, congrats

Author Response

                                                                REVIEW NO. 3 (highlighted in green)

Reviewer's recommendations

Authors’ Responses

Dear Authors:

Thanks by share your study.

I have some observations.

1. Introduction is extensive. There is an explain about the definition of mood, including the types. But all thats concepts are not  used in the rest of the paper, so I suggest summarizing.

The same goes for concept of emotion. For example, you present Plutchik's theory of emotions, where the basic emotions are listed, including disgust and trust. These emotions are not included in this way in the  instrument of evaluation that you later use.

We agree with the Reviewer and after thinking about it, we propose to modify the text and remove some excerpts of the text (highlighted in green).

2. The aim of the study is very broad. As a pilot study, I'm agree using this kind of objective. But you can give  to the readers a stronger arguments in the introduction about why you want to work in this topic. I suggest that clarify if  your focus is on engage, or work environment, well-being, patient safety, second victim, or another.

Text added: In the section "The aim of the study" (highlighted in green) "Mood and emotions motivate our behavior. Thanks to emotions, we establish relationships and function in the work environment. Making nurses more aware of the factors that determine their mood and emotions can help them cope with the demands of their occupational duties, promote their well-being and the quality of nursing care, including pediatric patient safety.”

3. When you use a methodology of investigation an instrument, it's good to explain why did you chosen a survey to explore the mood and emotions; and then, why did you chosen the Scale for measurements of mood and six emotions. Are there another methodologies and instruments? Why are the ones you chose correct?

Explanation for the Reviewer: The authors found the research tools by Bogdan Wojciszke and Wiesław Baryła were appropriate for the population of Polish pediatric nurses. It is a standardized research tool, understandable to the person participating in the study. It is characterized by high reliability and is used in the individual diagnosis and scientific research.

4. In line 184, when you explained how to get score of emotion questionnaire, wrote "by calculating the mean of the four detailed emotion components". Is not clear, because the instrument contain six emotions.

Corrected in the text: six

5. You obtained a lot of results, and you mentioned the relation of some variables selected with general mood scale, positive and negative, positive and negative mood scales and six emotions. Some results are surprising and need to give to readers some hypothesis that you can try to advance in discussion item. For example, why the people with specialization present more anger and guilty. Unless the sample was so small on this specific topic that the finding isn't really relevant. But this should also clarify it.

Explanation for the Reviewer: The authors admit that some of the results are surprising to them, but an additional possible explanation of the situation indicated by the Reviewer was introduced in the Discussion.

Text added: (in yellow) „In our study, higher education of pediatric nurses influenced the occurrence of positive emotions such as love, but having a pediatric specialty was also associated significantly more often with negative emotions such as anger and guilt. This was probably due to the incorrect system of work organization in the pediatric ward, including the overloading of pediatric nurses with duties, with the simultaneous lack of adequate financial gratification”.

6.I suggest including in the limitations a mention that the mood and emotions measured through a survey are affected by a specific local context (my service, my hospital) and national (my health system, my country, context), contingencies, and also cultural influences. Thus, the conclusions of this study cannot be universalized to other pediatric nurses in other countries.

Again, congrats

Additional text added to the Limitations section: „In addition, the mood and emotions of pediatric nurses measured in the survey using the aforementioned research tools have a specific local and national context related to the organization of the health care system, conditioned by cultural influences. Therefore, the conclusions of this study cannot be universalized for pediatric nurses in other countries”.

Round 2

Reviewer 1 Report

The paper has improved considerably. Almost all suggestions have been incorporated. Congratulations to the research team.

However, it must be explained if a pilot test was carried out with the questionnarie. 

Author Response

The authors of the article would like to thank the Reviewers for their valuable comments. The table below shows changes to the content of the article based on the suggestions provided.

The following part of the text has been added in the section 2.1 The corse of the study:

The research was preceded with the pilot study based on the questionnaire utilized in the authors’ own research. The pilot study encompassed 25 pediatric nurses. The results of the pilot study were included in the analysis of the study group.

Reviewer 2 Report

This manuscript can be accepted.

Author Response

The authors of the article would like to thank the Reviewers for their valuable comments. The table below shows changes to the content of the article based on the suggestions provided.

Another proof reading has been done to improve English in the article by a translator specializing in medical texts. Changes have been highlighted in grey. Minor lexical changes have been introduced. Decimals have been corrected writing a point instead of a comma. The abbreviations of the scales used have been corrected to correspond to their English names.